# Diurnal and Seasonal Variations in the Effect of Urban Environmental Factors on Air Temperature: A Consecutive Regression Analysis Approach

**DOI:** 10.3390/ijerph17020421

**Published:** 2020-01-08

**Authors:** Jaehyun Ha, Yeri Choi, Sugie Lee, Kyushik Oh

**Affiliations:** 1Department of Urban Planning and Engineering, Hanyang University, Seoul 04763, Korea; jaehyunha@hanyang.ac.kr (J.H.); ksoh@hanyang.ac.kr (K.O.); 2Department of Civil, Environmental and Geomatic Engineering, ETH Zurich, 8093 Zurich, Switzerland; yerchoi@ethz.ch

**Keywords:** air temperature, diurnal effects, seasonal effects, built environment, automatic weather station

## Abstract

This study investigates the diurnal and seasonal variations in the effect of environmental features on air temperature in Seoul, Korea. We expect that this study will lead to the identification of factors that can be applied for urban heat island mitigation strategies in summer without leading to an unintended result in winter. As our dependent variable, we employed the smoothed 31-day moving average of air temperatures, where we controlled the seasonal variation by normalizing the values observed from 247 automatic weather stations (AWS) from 2015 to 2016. Subsequently, we conducted consecutive log–log regression analyses of each day to examine patterns of change in regression coefficients and the significance of each independent variable. For independent variables, we applied built environment features including albedo, land-use, average building floors, the sky view factor, and green and water areas. This study provides analytical results regarding the relationship between environmental factors and air temperature. This study also addresses imperative issues for planners, especially regarding albedo, wind path, building geometry, and land use types. Finally, this study gives useful insights for managing the diurnal and seasonal variations of urban thermal environment in the mega-city.

## 1. Introduction

In recent decades, the urban heat island (UHI) effect has been among the most well-documented effects of urbanization along with climate change [1]. While there are no strict definitions of UHI, it is generally defined as a phenomenon in which the temperature difference between the city center and its suburbs is positive [2,3,4]. The UHI effect has been viewed as a severe threat to populations since high temperatures and heat waves may cause an increase in heat-related mortality [5]. In our study area, Seoul, the capital of South Korea, the number of heat-related illness reports has increased on average. From 2011 to 2015, 287 cases of heat-related illness were reported, an average of 57.4 per year. This increased to 170 in 2016, according to the Korea Centers for Disease Control and Prevention. Considering the growing senior population, who are at higher risk during heat waves, it is necessary for planners to mitigate the high urban temperature in Seoul.

Regarding the UHI effect, architectural fields have mainly discussed microscale variables such as building facades, construction materials, and the effects of cool roofs and vegetation in buildings [6,7]. In contrast, the planning and design fields have attempted to address macroscale variables such as land use patterns, built-up and vegetation index, traffic roads, and open water bodies [8]. The fundamental theories regarding the influence of building geometry on UHI were developed earlier by Nunez and Oke [9] and Oke [10], which has led to several empirical studies in recent years [9,10]. These studies focused on morphological indicators such as the sky view factor (SVF), urban porosity, and surface roughness [11,12,13]. Moreover, they suggested that the geometric configuration of buildings can affect the urban temperature both directly and indirectly by influencing other factors such as solar radiation and ventilation performances [14,15].

The latest studies have focused on the diurnal and seasonal variations in the impact of those previously mentioned parameters on urban temperature. Particularly, Chun and Guhathakurta [11] insisted that urban heat mitigation strategies should benefit over the diurnal cycle. Additionally, Yokobori and Ohta [16] reported that the UHI effect has a different diurnal pattern by season, and Chun and Guldmann [17] mentioned that strategies for ameliorating UHI should incorporate all seasons. However, these studies did not show consistent results for the determinants of urban temperature regarding the diurnal and seasonal pattern, enough to support and lead to devising comprehensive UHI mitigation strategies. Additionally, most studies have focused on the land surface temperature, which is inappropriate to reflect the temperature directly sensed by the people. On the other hand, studies that applied the air temperature as their dependent variable also exhibits a critical limitation due to an insufficient number of samples. For example, Yan et al. employed only 26 sample datasets while Yuan and Chen used 18 sample areas [18,19]. Thus, more robust and fine-tuned studies are needed to formulate pragmatic UHI mitigation tactics.

In this study, we investigated the diurnal and seasonal variations in the impact of urban environmental elements on air temperature. We expect that the results will lead to the identification of factors that can be employed in UHI mitigation strategies in summer, without causing adverse effects during winter. The study area is the mega-city of Seoul, South Korea, which is composed of various built environmental characteristics and presents weather features of both summer and winter. Meanwhile, we applied high-resolution climate data observed by 247 automatic weather stations (AWSs) for two consecutive years (2015 to 2016). Additionally, we employed the normalized value of the smoothed 31-day moving average of the air temperature as our dependent variable.

## 2. Literature Review: Effect of Urban Factors on Temperature

A number of studies on the impact of surface attributes on urban temperature have provided conforming results; for example, impervious surfaces were found to increase the accretion of solar radiation during the day and its ensuing release at night. Higher building footprints and lower vegetation levels were also reported to increase the temperature, although a few studies have shown opposite results. Chun and Guhathakurta [20] proposed that larger building roof surfaces were associated with lower temperature since roofs quickly release heat energy after sunset. They also reported that vegetation volume increases the surface temperature at night due to the trapped warm air under the canopies. Chun and Guldmann [17] showed that building footprints have a positive influence on daytime UHI during warmer seasons, which have longer daylight and steeper solar incident angles. Meanwhile, Yan et al. [18] suggested that the building area has a insignificant relationship with air temperature, and also found that vegetation area has a relationship with air temperature only during the nighttime. Additionally, Shiflett et al. [21] reported that the vegetation provides cooling effects, although the effect varies with time of day.

Cool roof and albedo levels are also widely used as indicators to explain urban temperature. Santamouris [6] reported that cool pavements have a significant potential for decreasing the temperature, both ambient and surface, based on a literature review relevant to cool pavement design strategies. Furthermore, a recent study by Wang et al. [22] showed that applying cool pavement strategies for alleviating UHI worked better in middle-rise areas than in high-rise areas and attributed this result to the long distance from the surface to the cool pavement in high-rise areas. Taha [23] undertook one of the earliest studies that explained the impact of albedo on the surface and near-surface air temperature. Moreover, this relationship between albedo and temperature has been confirmed in several studies [24,25,26], implying reliable indications for UHI mitigation.

Although several studies have provided policy implications for UHI mitigation regarding albedo, recent studies have raised issues that should be further investigated [26,27]. For example, Yang et al. [27] provided a synthetic overview on reflective materials and argued that further research should provide empirical results from a seasonal perspective. Their exposition reveals the necessity of understanding how albedo is associated with the temperature in winter, since lower temperatures resulting from a higher albedo may cause side-effects during cold seasons. Additionally, Liu et al. [26] reported a positive relationship between albedo and land surface temperature during the daytime and attributed this result to water bodies, explaining that water bodies are composed of low albedo values while exhibiting cooling effects. Furthermore, Hamada and Ohta [28] showed that higher vegetation decreased the air temperature in summer, while the effect was negligible in winter. They attributed this phenomenon to the deciduous trees, which generate fewer shading effects in winter, implying green space as a practical design tool that can work throughout the season.

The influence of land use type on urban temperature has also been investigated. Jusuf et al. [29] examined the impact of land use type on both daytime and nighttime surface temperatures. They observed that industrial and commercial airport areas had higher daytime temperatures compared to other areas. In addition, Bokaie et al. [30] investigated the relationship between seasonal change in land surface temperature and land use/land cover (LULC) classes in Tehran in 2010 and concluded that UHI effects had different causes due to the type of LULC classes in the region. Similarly, Van and Bao [31] reported that the highest surface temperature cores were in industrial and urban areas, and land cover composed of forest or water showed lower temperatures. Hart and Sailor [32] showed that the area with the warmest air temperature was a locus associated with industrial use, possibly due to a lack of vegetation and constant anthropogenic heat release. They also found that UHI intensity in the downtown core was smaller, probably due to the shade created by the high-rise buildings, which is in line with Zhang et al. [33]. Unlike previous studies, Krüger and Givoni [34] reported that the relationships between land use and local climate were uncertain and further emphasized that relating land use features to local temperatures is a difficult challenge. 

Aligned with the land use indicators, areas of built-up areas and traffic roads, green spaces, and water bodies were also examined [35]. Chun and Guhathakurta [20] reported that traffic road surface had a negative effect on surface temperature, and they exposited this relationship by demonstrating the air ventilation effect induced in open spaces. They also explained that much of the road surface was shadowed by street trees and adjacent buildings during the day. Correspondingly, Hart and Sailor [32] found that air temperature near arterial road surfaces was among the warmest due to low albedo. Kim and Kim [36] also showed that road use increased the air temperature regardless of season or time of day. On the other hand, a number of studies agreed that a higher fraction of green spaces and water bodies decreased the urban temperature [37,38]. Moreover, some studies have further fine-tuned the green space variable by applying park vegetation, shape, and gravity indices and examining the local and regional cooling effects of green spaces [39,40,41].

Another substantive part of previous studies regarding urban temperature is building geometry. Theoretically, urban geometry regulates the absorption and storage of incoming solar radiation and impedes wind ventilation, leading to UHI in densely built areas [2,9,10]. The impact of building geometry on temperature is intricated as dense building decreases the temperature through the shading effect, but increases the temperature due to the release of heat energy during nighttime [42,43]. Regarding this relationship, Lan and Zhan [43] suggested that increasing the building height and density may reduce the daytime air temperature, but lead to a significantly higher temperature at night. Additionally, the impacts of building volume or height variations were examined by Chun and Guhathakurta [20], and their results indicated that higher values of those two measures were associated with lower daytime surface temperature. Additionally, Yue et al. [44] showed that urban configuration characteristics, for example, the patch index and contiguity index, are one of the driving factors of UHI.

Moreover, several studies have applied the SVF index to explain air temperature. Theoretically, when SVF is low, the deep canyons reduce heat loss and decrease the longwave radiation from canyons, resulting in higher temperatures [10]. This notion is well supported by empirical studies [19,45]. Yuan and Chen [45] reported that a 10% increase in SVF could decrease the air temperature by about 0.4 °C between 2 and 4 a.m. They further implied that planners must consider decreasing the building density because it is difficult to increase the SVF in dense urban areas. Similarly, Chun and Guhathakurta [11,20] reported a negative coefficient of SVF during the day, explaining that more open space dissipates hot air in the same way as at night. However, contradictory results were also reported by other studies which showed that SVF was positively related to temperature, regardless of time [22,43,46,47]. Lan and Zhan [43] exposited these results in that lower SVF creates more building shade, lowering the diurnal air temperature.

Previous studies had several limitations regarding the relationship between SVF and urban temperature. The first is the sampling issue. For example, Bourbia and Boucheriba [48] reported a positive relationship between SVF and air temperature, while limiting their results to urban streets. When limited to urban streets, open areas are extremely exposed to the sun during the day and the dark asphalt induces a heat trap. Additionally, open areas in urban streets are likely to suffer from a lack of vegetation [48]. In addition, Cai et al. [39] demonstrated that the results may show discrepancies according to whether there are water bodies near the study area. The second limitation is about considering the wind as an explanatory variable. Eliasson and Svensson [49] suggested that there might be no correlation between air temperature and SVF in windy weather due to the mixed air mass. The third is the consideration of anthropogenic heat released from adjacent buildings. Yang et al. [46] proposed that the relationship between SVF and UHI intensity may be underestimated in situations where anthropogenic heat emissions are substantial. The last limitation is the different measurement methods of SVF. While using the fish-eye lens is the most conventional method, several other methods such as aggregated value, ArcView program, or Google Street View images have also been used [19,50,51]. Regarding this issue, Unger [52] suggested that the application of the areal average of SVF is more desirable when analyzing the relationship between temperature and SVF.

Meanwhile, recent studies have focused on understanding the determinants of urban temperature from a seasonal perspective. Furthermore, few studies have focused on analyzing the temporal trends of both surface urban heat islands and the intensity of land surface temperature [53,54,55]. Chun and Guldmann [17] used the bimonthly Landsat TM images to investigate the relationship between SVF and surface temperature; they reported that SVF had a negative impact on surface temperature consistently. In contrast, Yan et al. [18] showed that a higher SVF was related to higher daytime air temperature for both summer and winter. This result is more in line with the perspective that the shadow created by buildings, which indicates lower SVF, mitigates the daytime temperature. Their findings also indicate that SVF is negatively associated with the nighttime air temperature in winter. Besides, Tong et al. [56] reported that the SVF is negatively related to both daytime and nighttime air temperature during summer and only with nighttime temperature in winter.

The following are significant issues that should be addressed when analyzing the determinants of temperature from a diurnal and seasonal perspective. First, deciding the periods for daytime and nighttime is a significant challenge. For example, Chun and Guhathakurta [20] designated nighttime as 03:43 a.m. and daytime as 4:20 p.m. by using the Advanced Spaceborne Thermal Emission and Reflection Radiometer(ASTER) satellite image dataset. On the other hand, when applying air temperature, the period for daytime and nighttime varied by research. Yan et al. [18] selected 2 to 3 p.m. as daytime and 11 p.m. to 12 a.m. as nighttime, while Yokobori and Ohta [16] defined daytime and nighttime based on sunrise and sunset. Second, selecting a time period for each season is also challenging. For example, Park et al. [13] designated spring, summer, fall, and winter as April, August, October, and January, respectively. Notably, a few studies have divided the season based on climate conditions or by examining whether the average temperature is higher or lower than a certain degree [18,56]. Finally, when analyzed from a seasonal perspective, applying the changes in the independent variables is also challenging. Chun and Guldmann [17] measured the Normalized Difference Vegetation Index(NDVI) bimonthly to understand the seasonal pattern. However, NDVI values are difficult to calculate when applying daily weather data instead of satellite images.

To summarize, the diurnal and seasonal variations in the relationship between the built environment and urban temperature are ambiguous and yet to be clarified, but still intriguing. This abstruseness can be resolved by addressing the following issues: measuring air temperature by stationary AWS, employing a sufficient number of samples, and exploring seasonal variations by conducting continuous regression analysis. In particular, a daily and consecutive analysis is essential to explore the seasonal variations of the determinants of urban temperature. Moreover, to resolve the issues addressed in the previous section, the air temperature must be consecutively measured by fixed weather stations. By keeping these related issues in mind, it would be possible to provide consonant results to alleviate various problems in the urban thermal environment.

## 3. Methods

### 3.1. Study Area

The study area was Seoul, South Korea, comprising 605.3 km^2^ of land with a population of 9.8 million (Figure 1). Seoul is composed of various built environments including low semi-detached housing, mid-rise residential and commercial areas, and high-rise apartments and business complexes. According to a report by the Korea Meteorological Administration [57], the mean annual air temperature has increased by about 1.2 °C over the last 30 years. Seoul is also currently experiencing severe UHI effects, particularly in the densely built city center [58]. The average intensity of UHI in Seoul is approximately 1.8 °C, while the maximum intensity is about 4.3 °C [59]. The climate of Seoul is classified as ‘Dwa’ according to the Köppen classification, indicating cold, dry winters and hot summers [60]. This classification implies that Seoul has four distinct seasons, showing daily mean temperatures of 25.2–33.3 °C during August and −7.3–0.1 °C during January.

### 3.2. Dependent Variable

The dependent variable of this study was the air temperature measured from 247 AWSs located in Seoul including the peripheral areas (Figure 1). To investigate changes in the air temperature determinants, we used climate data collected from January 2015, to December 2016. We used datasets provided by the Korea Meteorological Administration (34), Seoul Metropolitan Government (25), and SK Weather Planet (198). The AWS equipment can measure air temperatures from −40–60 °C with an accuracy of ±0.3 °C. We also retrieved the wind speed data, measured within the range of 0–75 m/s with an accuracy of ±0.3%. The AWSs were installed on rooftops, about 1.5 m above the building roof. The rationale for this location was to avoid the effects of anthropogenic heat from automobiles and the building as well as to supply electricity. Meanwhile, Oke [61] clearly addressed that the air temperature measured from the roofs of buildings may be biased regarding the roofs’ materials and the airflows created by the flat-topped buildings. Oke [61], however, conceded that a meaningful air temperature could be measured when the AWSs are above the building roof by more than 1.5 meter, so that the roof and canyon air can be mixed. Therefore, although the data applied in our study may not be perfect to address the urban climate at a micro-scale level, it is still sufficient to analyze the air temperature in a local-scale level.

We used a smoothed 31-day moving average for climate variables. The moving average air temperature for 15 March 2015, was measured by calculating the average air temperature from 28 February to 30 March 2015. In particular, because the mean and standard deviation values of air temperature vary throughout the year, we normalized the dependent variable by subtracting each temperature value with the minimum temperature of each day. Additionally, we divided the values with the standard deviation value of each time period. To determine the diurnal variation, we used the average temperature for both daytime and nighttime. Daytime and nighttime were defined by the sun position. Daytime was designated as the first three hours after the time when the sun was at its meridian altitude, and nighttime was designated as the three hours after three hours since sunset. These methods were applied for each day. For instance, when the sun was at its meridian altitude at 12:36 (GMT + 9) and the sunset was at 18:28 (GMT + 9), daytime was set as 12:36–15:35 (GMT + 9) while nighttime was set as 21:28–24:27 (GMT + 9). The reason for applying the smoothed 31-day moving average was to obtain a clear pattern on the impacts of built environmental factors on air temperature.

### 3.3. Independent Variables

Our independent variables included the altitude of the AWS, average wind speed, solar radiation, surface albedo, land use factors, building geometry related measures, and natural resources. These had fixed values during our study period (2015–2016), with the exception of wind speed, solar radiation, and surface albedo. This is because other factors such as land use and building geometry were unfeasible to measure during the time scope due to data collection difficulties. Moreover, with regard to the air temperature, which is the dependent variable, it responds to the local-scale urban climate, so we measured the independent variables based on the 500-m circular buffer. The average wind speed variable was calculated from the same dataset used for the air temperature value, while also applying the 31-day moving average concept as addressed above.

Next, to measure solar radiation and surface albedo, we created a 500-m circular buffer for each AWS location. We employed the Area Solar Radiation tool of ArcGIS Pro software to compute the solar radiation. Based on the raster file, which consisted of the digital elevation model (DEM) and building geometry, we iterated the Area Solar Radiation tool 731 times and calculated the total solar radiation absorbed for each day within the circular buffer. We also applied the 31-day moving average concept. It should be noted that the solar radiation values may have a discrepancy with the actual solar radiation since the atmospheric conditions such as aerosols or clouds were not considered when the values were computed. However, the values are expected to well present the potential solar radiation based on the digital elevation model and geometry environments. For surface albedo, we used ten image sets captured during 2015 and 2016 from the Landsat 8 dataset. Then, we averaged the values for each circular buffer centered on each AWS, which indicated the average surface albedo value during 2015 and 2016.

We calculated the gross floor area of buildings located within the 500-meter buffer area to measure the land use factors. We classified land use types as residential, commercial, business/industrial, and traffic road. Business and industrial use were combined because Seoul’s industrial areas are adequate to consider them as a business rather than industrial use. We used both the building ledger data and the building shape data provided by the National Spatial Data Infrastructure Portal of Korea. In addition, we used the ecology data retrieved from the National Environment Information Network System of Korea to measure traffic road and natural resources. We classified natural resources into artificial green space, natural green space, and water space. In detail, the natural green area are the green areas formed by topography such as hills or mountains, while the artificial green area refers to small and large urban parks created artificially. Each variable was measured by the area within the 500-m circular buffer.

Finally, we calculated the average building floors and SVF values to consider the building size and configuration. To measure the average building floors within the 500-m circular buffer, we used the building ledger data. The SVF, the portion of the visible sky at a certain point, was measured by the average value within the buffer. It has been reported that the areal SVF can better address the air temperature rather than the SVF measured at a certain point [52]. To do so, we used the 1 m x 1 m raster data, forged by merging the DEM and the building floor data. Moreover, since the air temperature variable was observed from the building roofs where the roof and canyon air are mixed, we deemed that the SVF values should represent the areal average.

### 3.4. Methodology

Descriptive statistics for both the dependent and independent measures were first examined. Since variables such as air temperature, wind speed, and solar radiation fluctuate across time, we examined their changes from 2015 to 2016. As a next step, we conducted regression analysis 1402 times in order to explore the diurnal and seasonal variations in the impact of environmental factors on the air temperature. There were 701 days in our research period, resulting in 1402 regression models, considering both daytime and nighttime.

In our model, both the dependent and independent variables were transformed into log form. The rationale of this log–log regression approach was to fine-tune the data, reduce any unintended effects of outliers, and interpret the regression coefficients as elasticities. In more detail, the coefficients estimated by the log–log regression model are invariant to the scales of variables because they measure the expected percentage change in the dependent variable when the independent variable increases by one percent. Interpreting the coefficient as an elasticity makes it feasible to compare the influence of a specific variable across time and other variables. We both interpreted and visualized the changing patterns in r-square values, regression coefficients, and its significance level throughout the two years. We also checked the variance inflation factor (VIF) to avoid biased results.

## 4. Results

### 4.1. Descriptive Analysis

Figure 2 shows the diurnal and seasonal variations of the air temperature, which is the raw data of our dependent variable. In detail, the air temperature was spatially averaged for both daytime and nighttime. The highest air temperatures of both daytime and nighttime were measured between July and August, while the lowest air temperatures were recorded around 15 January 2016.

It should also be noted that the trend of air temperature was irregular in September 2015. It is possible to see that the changes in air temperature across time had significantly decreased, and it showed a pattern that was not observed in other periods. The abnormal pattern may be attributed to the significant changes in air temperature that occurred before and after 15 September 2015; the daily average air temperature significantly dropped from 26.3 °C to 21.7 °C on 25 and 26 August 2015, while it also changed from 21.3 °C to 16.3 °C on 1 and 2 October 2015.

Figure 3 shows the diurnal and seasonal variations of the average wind speed, which was one of our independent variables. A significant pattern for daytime was apparent: the average wind speeds were high around 15 February in both years and showed the lowest values around November and December. The nighttime wind speed did not exhibit a clear changing pattern, although some fluctuations could be observed.

Table 1 shows the descriptive statistics for the independent variables. We used log–transformation for all of these variables to conduct a log–log regression analysis. The average altitude of the AWS equipment was 63.1 m. Although most AWSs were located on the top of three to four-story buildings, the altitude of AWSs was highly varied due to the DEM values. For the surface albedo, the value represents the average within the 500-m circular buffer. Regarding land use attributes, the variables represent the gross floor area. For example, the maximum value of the residential use was 1.478, implying that the floor area ratio was around 1.88. The average SVF value was 0.63, while the lowest SVF value was measured in the Gangnam area, one of the city centers. For green and water bodies, the standard deviation values were relatively large because most of the 500-m circular buffers did not contain green and water bodies. Additionally, Table 1 shows the variance inflation factor (VIF) for the independent variables related to the built environment.

Table 2 shows the correlation matrix of the independent variables applied in our analysis. Although the variables do not have a multicollinearity issue based on the VIF values, some variables showed high correlation. For instance, the floor areas of commercial use and business use showed a high correlation of 0.664. In addition, the areal sky view factor was correlated with both the artificial and natural green area. Moreover, the areal sky view factor showed a negative correlation with the floor area of each land use type while showing a positive relationship with green or water area.

### 4.2. Consecutive Regression Analysis

In this section, we examine the diurnal and the seasonal pattern of how the environmental factors affect the 31-day moving average of air temperature, which we normalized to control the seasonal variation. Figure 4 shows the R-square value of 1402 regression models, which was based on all of the independent variables addressed in the previous section. As seen in Figure 4, the R-square for daytime models were between 0.3 and 0.5, while it varied across the two years. On the other hand, the R-square for the nighttime models was between 0.6 and 0.7 and showed less fluctuation. Based on these results, we can infer that the environmental factors better explain the nighttime air temperature than daytime, further implying the main reasons of the UHI formation. Moreover, around 15 September 2015, the R-square significantly declined, and this result can be attributed to the notable temperature change addressed previously. Figure 5, Figure 6, Figure 7, Figure 8, Figure 9, Figure 10, Figure 11, Figure 12, Figure 13, Figure 14, Figure 15, Figure 16 and Figure 17 show the diurnal and seasonal patterns of the independent variables. It should be noted that the range of the y-axis varied with each independent variable because the range of coefficients differed by variable.

Figure 5 shows the changing patterns in the coefficients of the altitude of the AWS. Here, the y-axis refers to the coefficient values based on the log–log regression, which can be both interpreted as elasticity and coefficient. As seen in Figure 5, the influence of the altitude on daytime and nighttime air temperature was analogous. Regarding the seasonal pattern, the coefficient values of the log–log regression models were between −0.02 and −0.03, while they were all statistically significant. The coefficients were slightly lower during the summer seasons compared to other seasons. This result may be due to the different temperature laps rate by season, further showing a significant difference between summer and winter.

The diurnal and seasonal variances of the coefficients of average wind speed are addressed in Figure 6. For daytime, a higher wind speed resulted in lower air temperature, while the intensity of the influence was slightly larger during the spring. For example, a 10% increase in daytime wind speed would result in a −0.6% change of air temperature in summer and −0.3% in winter. It is interesting that the patterns of the coefficients for nighttime were positive as well as stable. This result could be explained by the discrepancy of the wind direction between daytime and nighttime. Furthermore, it implies that creating a wind path to accelerate wind speed may cause adverse effects such as increasing the nighttime air temperature during all seasons and decreasing the daytime air temperature in winter, both of which are undesirable.

Next, the influence of solar radiation on air temperature was negligible. As seen in Figure 7, the coefficients were mostly not significant. Exceptionably, the coefficients were intermittently significant in August and September of 2015. However, it is inappropriate to interpret these results because the dependent variables in this period showed extreme fluctuations due to drastic weather change. It seems plausible that the effect of solar radiation on air temperature is consummately controlled by other variables such as albedo and building geometry. Moreover, although the solar radiation variable may not represent the precise amount of radiation, it is possible to argue that the effect of solar radiation on air temperature is insignificant.

As shown in Figure 8, the relationship between surface albedo and the air temperature was mostly significant for both the daytime and nighttime. For daytime, the results indicated that an increase in surface albedo would reduce the air temperature; however, the coefficients were not significant during the winter seasons. On the other hand, the coefficients of surface albedo during nighttime were significant regardless of the seasonal time period. The coefficients showed a seasonal fluctuation, while the absolute values were higher during the spring seasons. Based on this result, it is plausible to conclude that increasing the surface albedo could help mitigate UHI. However, planners must note that altering the surface albedo to mitigate the urban air temperature may result in unintended outcomes. For example, an increase in surface albedo would decrease the nighttime air temperature, especially during spring and winter, which may further lead to more energy use. Therefore, policymakers should keep in mind that considering albedo as a design factor to alleviate air temperature may cause unintended outcomes.

The diurnal and seasonal patterns of the relationship between the residential area and air temperature are shown in Figure 9. The association between the two variables at daytime and nighttime were similar, while its influence was mostly higher during nighttime. In a seasonal perspective, the relationship was not significant for some time periods, especially in nighttime from late summer to fall. Still, it is possible to conclude that the relationship between residential use and the air temperature was positive, while the coefficients were greater during nighttime. This result may be attributed to the anthropogenic heat that occurred from the residential buildings after daytime hours; however, further investigation may help better understand this relationship.

As shown in Figure 10, the patterns of the coefficients of commercial area and air temperature showed a significant discrepancy between daytime and nighttime. For instance, during the daytime, the coefficients of the commercial area variable showed a negative sign, while the coefficients were significant only during winter. This result implies that the areas with high commercial use might experience a cool island during the daytime in winter seasons. On the other hand, the coefficients of the commercial area during nighttime were positive, showing robust significance during most seasons. Regarding the importance of mitigating the air temperature, it is possible to infer that the air temperature in commercial areas are likely to be high between the spring seasons during the nighttime. It is also possible to understand that the effect of commercial area on air temperature has perceivable diurnal and seasonal variations.

As shown in Figure 11, the effect of business and industrial areas on air temperature were mostly significant from late fall to winter. Remarkably, the effect of business and industrial areas on air temperature was relatively higher during nighttime. For instance, considering the coefficient values, it is possible to say that a 10% increase in business and industrial areas may lead to a 0.06% increase in air temperature in winter nighttime. Although the absolute values of the coefficients were small, it is plausible to understand that business and industrial areas are one of the contributors of high air temperature, in terms of land use. Besides, as seen in Figure 9, Figure 10 and Figure 11, the effect of land use on air temperature showed significantly different patterns between residential, commercial, and business and industrial use. Based on these results, we could account for areas where air temperatures would be high while regarding both diurnal and seasonal patterns

Our results showed that larger traffic road area is associated with higher air temperature throughout all seasons except for winter daytime (see Figure 12). The coefficients were higher during nighttime than daytime, indicating that the main materials of traffic road are likely to release the heat absorbed during the daytime, resulting in higher nocturnal air temperature. It should also be noted that the pattern of the coefficients during 2016 is more reliable, keeping in mind that the dependent variables of 15 September 2015, were not consistent. This result ascertains that a larger traffic road area increases the air temperature more in the nighttime during the summer seasons.

As shown in Figure 13, the building floor average, which can describe the building geometry, did not show a significant relationship with the daytime air temperature. This could be explained by the fact that higher building floors not only produce and absorb more heat but also create shade that lowers the daytime air temperature of both the roofs and canyons. The influence of the building floor average on nighttime air temperature showed a positive sign. However, the significances of the coefficients during spring and early summer were not significant, especially in 2015. Additionally, the coefficients of the building floor average were slightly higher during fall compared to other seasons. This indicates that areas with higher building floors may show significant nocturnal UHI. Based on these results, it seems that the effect of the building floor average on daytime air temperature should be further investigated.

SVF was shown to be less associated with the daytime air temperature throughout the season (Figure 14). In detail, the coefficients of the SVF were positive during the daytime and were statistically significant only during spring. However, the values of the coefficients were small. For instance, a 10% increase in SVF would lead to an approximately 0.3% increase in daytime air temperature. This result is in line with previous studies that presumed that a higher SVF would lead to more solar radiation absorption [22,43,46,47]. On the other hand, the coefficients of the SVF for nighttime air temperature models showed a negative sign. The coefficients were mostly significant, except from winter to spring. Based on this result, it is possible to argue that increasing the SVF would help mitigate the nocturnal air temperature. This result further implies that policymakers should give more attention to SVF since increasing the SVF value would lower nocturnal air temperature during summer without leading to any other unintended outcomes.

Figure 15 shows the changing patterns in the regression coefficients and their significances for the artificial green area. As the blue dots show, more artificial green areas would reduce the daytime air temperature. In more detail, the results were mostly statistically significant for spring and summer, while the absolute values of coefficients were large. On the other hand, the relationship between the artificial green area and nocturnal air temperature were insignificant regardless of the season. This result requires further investigation because several previous studies have pointed out that green space would reduce and mitigate the air temperature as well as the UHI effect. Therefore, creating artificial green areas with an expectation of a decline in air temperature may be limited to daytime during spring and summer seasons.

Figure 16 shows the diurnal and seasonal variations in the effect of natural green area on air temperature. As indicated by the blue dots, larger natural green area was associated with lower daytime air temperature. However, the coefficients were significant only during spring and winter. For instance, during spring, a 10% increase in the natural green area would reduce the daytime air temperature by about 0.1%. This result is similar to the artificial green area as shown in Figure 15. On the other hand, the natural green area did not show a statistically significant association with nighttime air temperature. To explain this result, it is possible that the natural green area within the circular buffer did not influence the air temperature of a point. Instead, natural green area may influence air temperature on either relatively smaller or broader scales.

Finally, Figure 17 shows the changing patterns in the regression coefficients of the water area. The results indicate that water bodies may increase the daytime and nighttime air temperature occasionally. Especially, water bodies were significantly associated with the daytime air temperature during spring and fall. On the other hand, the results for nighttime air temperature were not robust. As seen in the figure, a larger amount of water was associated with higher nighttime air temperature, however, the relationship was not always significant. However, this result may not apply to other areas because most of the samples were far away from large water bodies (see Figure 1).

## 5. Discussion

This study focused on the diurnal and seasonal patterns of determinant variables on air temperature, which was the normalized value of the 31-day moving average air temperature. Based on the analysis, the main points of this study are as follows. First, while the R-square value for the daytime models were approximately 0.3–0.5, the R-square for nighttime increased to 0.6–0.7, indicating that the urban environmental factors better explained the nocturnal air temperature than the daytime one. This result further implies that the UHI effect, which is known to occur after sunset, can be attributed to the built environmental characteristics. From a seasonal perspective, the daytime temperatures were well explained, especially for the summer seasons when compared to winter. Thus, it is probable that the daytime air temperature in summer could be managed by planning and design strategies.

Second, we examined the effects of altitude, wind speed, and solar radiation on diurnal and seasonal air temperature. The air temperature was negatively associated with altitude, regardless of time, while solar radiation did not show a statistically significant relationship. This implies that solar radiation levels do not have much influence on the air temperature when other environmental features are controlled. Higher wind speed tended to decrease the daytime air temperature regardless of season while increasing the nocturnal air temperature. This indicates that creating wind paths to mitigate air temperature may produce adverse effects such as lowering the daytime air temperature in winter. Thus, planners should be aware of wind directions when creating wind paths in order to avoid any unintended air temperature changes caused by wind.

Third, albedo showed a negative relationship with diurnal air temperature for most seasons. However, the results also indicated that we should be aware of its effect during cold seasons. This is because higher albedo decreases the air temperature during winter, which is not desirable. Thus, we suggest that planners and policymakers, especially those from urban areas that have both hot and cold seasons, should be aware that altering the albedo to mitigate air temperature may lead to unwanted outcomes. We acknowledge the cooling effect of higher surface albedo, but insist that using albedo for UHI mitigation is more appropriate in tropical cities such as Hong Kong or Singapore where there is less cold weather. On the other hand, readers should note that higher albedo may rather increase the apparent temperature and heat stress, which considers wind speed and radiation as well as humidity [62,63]. In this regard, future studies may focus on how albedo impacts either air temperature or apparent temperature from a seasonal perspective.

Fourth, the impact of land use variables on air temperature differed by land use types. Taking account of nighttime air temperature, commercial and business/industrial areas may be significant determinants of air temperature throughout the year, while residential areas are the main factors during spring and summer. According to our results, it is also possible to perceive how the impact of each land use type on air temperature changes. Thus, our results are expected to help policymakers understand the convoluted relationship between land use variables and air temperature. Still, future studies may try to reveal the innated relationship between land use type and air temperature from the aspect of anthropogenic heat occurred by energy use.

Finally, based on the coefficients of the building floor area and the SVF, the following implications were derived. The average building floor did not have a significant effect on daytime air temperature. This could be attributed to the ambivalent characteristics of the average building floor: higher building floors might trap more heat, but also create more shade during the daytime. In fact, although it is generally known that higher building height may hamper air circulation and trap heat [64], empirical studies suggest that higher buildings may reduce the solar radiation absorbed by the ground and building facades [20]. The SVF variable mostly showed an insignificant relationship with daytime air temperature, but a significant negative association with nighttime air temperature during spring and fall seasons. The insignificant relationship between the SVF and daytime air temperature can be explained by the ambivalent effect of a high SVF: higher solar radiation leads to the increase of air temperature while inducing air circulation, which decreases air temperature.

## 6. Conclusions

This study investigated the diurnal and seasonal variations in the impact of environmental features on the normalized value of the 31-day moving average air temperatures. We derived implications by examining the changing patterns in regression coefficients and their significances by iterating regression analysis. The main focus of this study was to contribute to examining the determinants of air temperature from both the diurnal and seasonal perspectives.

This study provides useful results regarding the relationship between environmental factors and air temperature. Our analysis offers insights into managing the thermal environment throughout both the day and year. Additionally, this paper addresses imperative issues that planners should acknowledge, especially regarding albedo, wind path, building geometry, and land use types. Further studies may still expand this research in the following aspects: first, future studies may fine-tune the independent variable so that it considers the seasonal change. For instance, changes in vegetation could be accounted for. Second, we suggest that future studies investigate the seasonal changes in anthropogenic heat by land use type and relate it with the air temperature.

Despite the contribution of this paper, some limitations should also be highlighted. First of all, the air temperature measured from the rooftop level may not well represent the temperature at a local scale. For instance, the air temperature applied in this paper may be affected by the heat released from the building or the material type of the roof. However, this case would also apply to the air temperature measured at a street level. Second, the albedo variable was measured by using the Landsat image, which represents the average albedo value of the building rooftops as well as the streets. We suggest that future works should focus more on how albedo affects air temperature by its type from a seasonal perspective. Finally, we did not control for the spatial autocorrelation issue that might be present in our dependent variable. This is because the level of spatial autocorrelation may differ throughout the temporal scope of our research, making it difficult to understand the seasonal pattern of the coefficients of other independent variables. Future studies could focus on addressing the spatial autocorrelation issue while analyzing the determinants of air temperature from a seasonal perspective.

## Figures and Tables

**Figure 1 ijerph-17-00421-f001:**
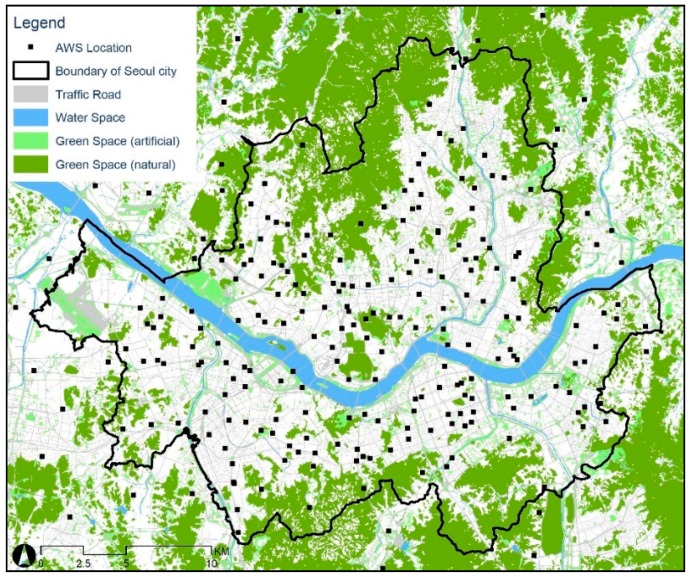
Study area.

**Figure 2 ijerph-17-00421-f002:**
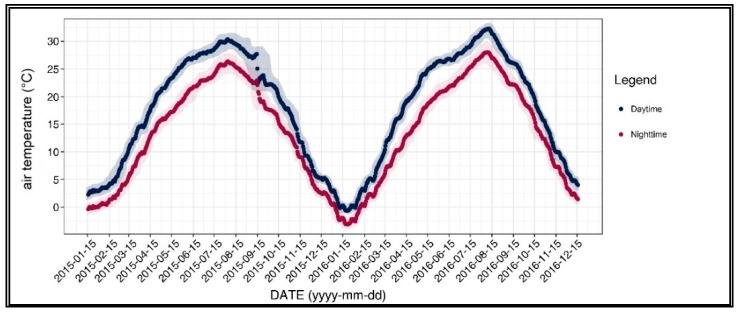
Air temperature of the 31-day moving average.

**Figure 3 ijerph-17-00421-f003:**
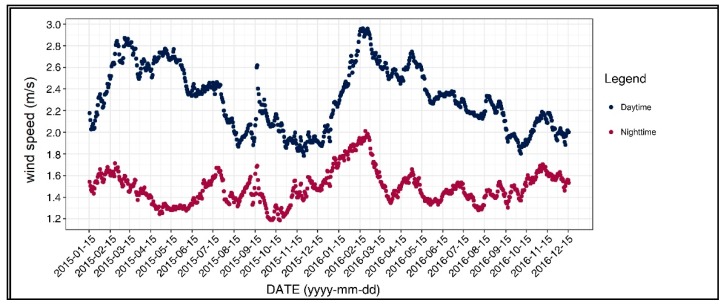
Wind speed of the 31-day moving average.

**Figure 4 ijerph-17-00421-f004:**
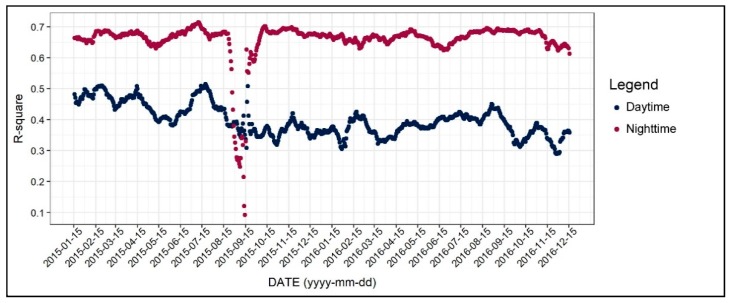
Changing patterns in the R-square value.

**Figure 5 ijerph-17-00421-f005:**
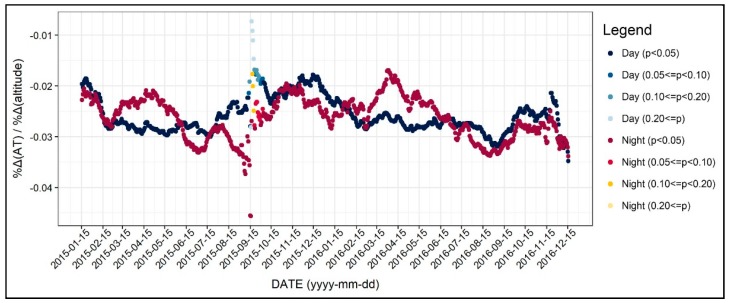
Changing patterns in the regression coefficients and their significances (Altutude of the automatic weather station (AWS)).

**Figure 6 ijerph-17-00421-f006:**
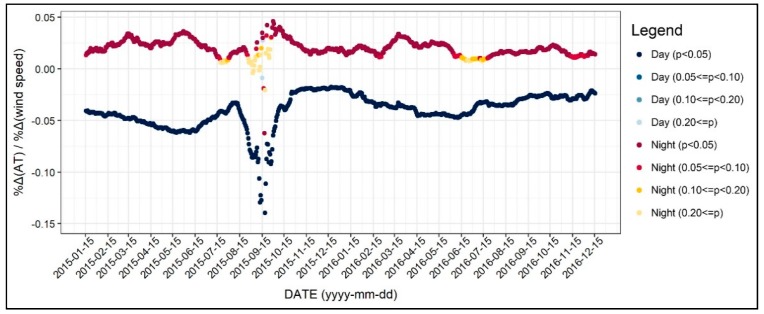
Changing patterns in the regression coefficients and their significances (wind speed).

**Figure 7 ijerph-17-00421-f007:**
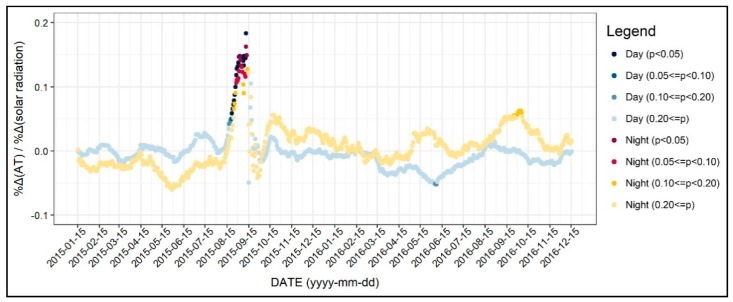
Changing patterns in the regression coefficients and their significances (solar radiation).

**Figure 8 ijerph-17-00421-f008:**
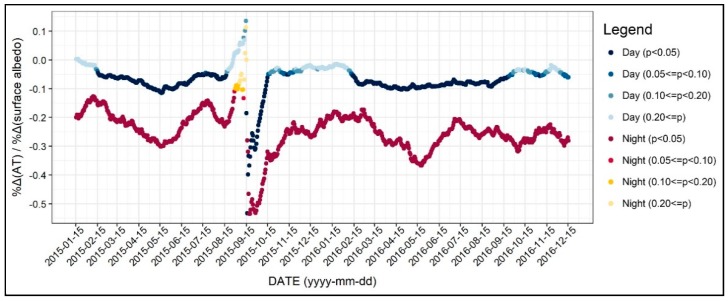
Changing patterns in the regression coefficients and their significances (surface albedo).

**Figure 9 ijerph-17-00421-f009:**
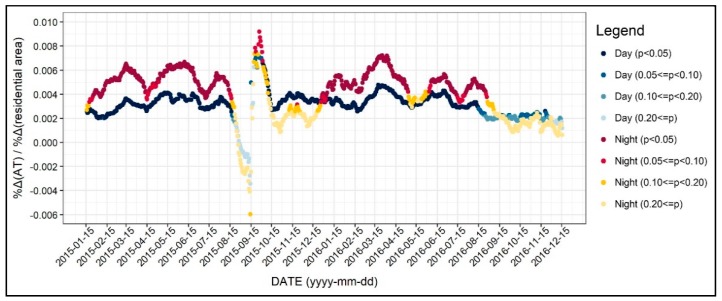
Changing patterns in the regression coefficients and their significances (residential use).

**Figure 10 ijerph-17-00421-f010:**
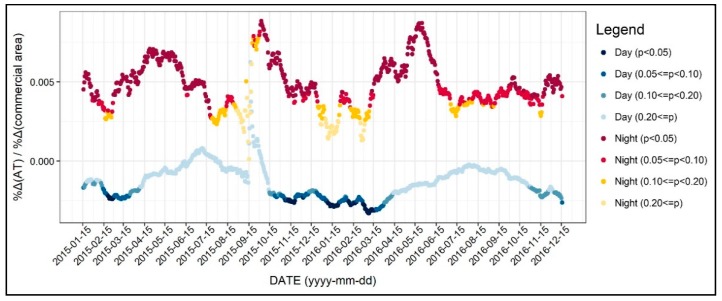
Changing patterns in the regression coefficients and their significances (commercial use).

**Figure 11 ijerph-17-00421-f011:**
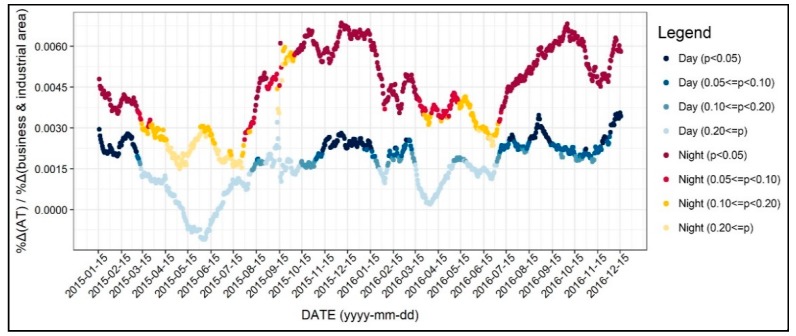
Changing patterns in the regression coefficients and their significances (business and industrial use).

**Figure 12 ijerph-17-00421-f012:**
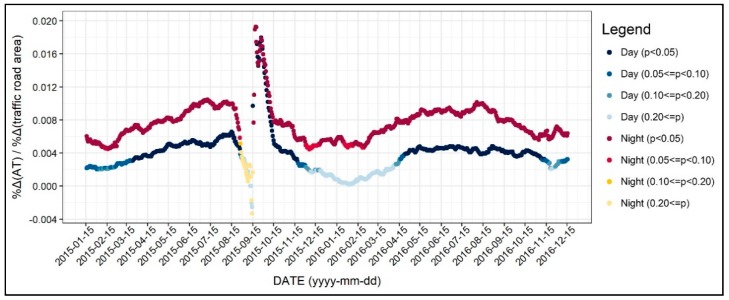
Changing patterns in the regression coefficients and their significances (traffic road area).

**Figure 13 ijerph-17-00421-f013:**
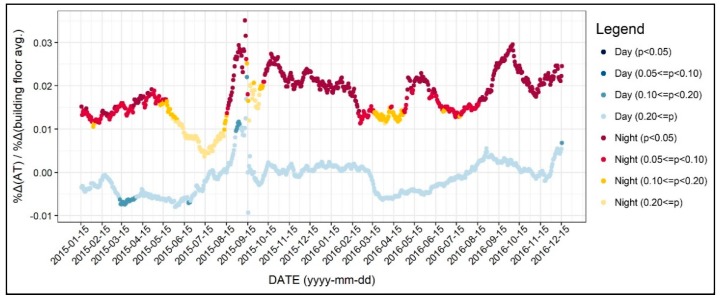
Changing patterns in the regression coefficients and their significances (building floor avg.).

**Figure 14 ijerph-17-00421-f014:**
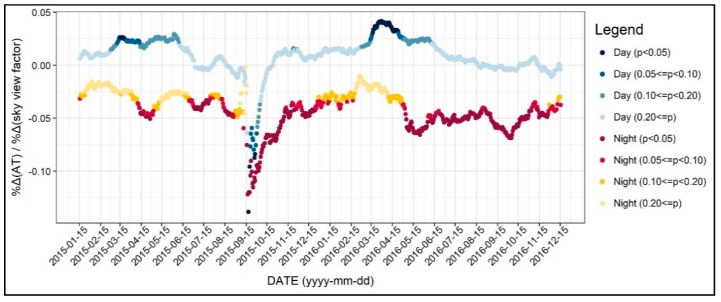
Changing patterns in the regression coefficients and their significances (sky view factor).

**Figure 15 ijerph-17-00421-f015:**
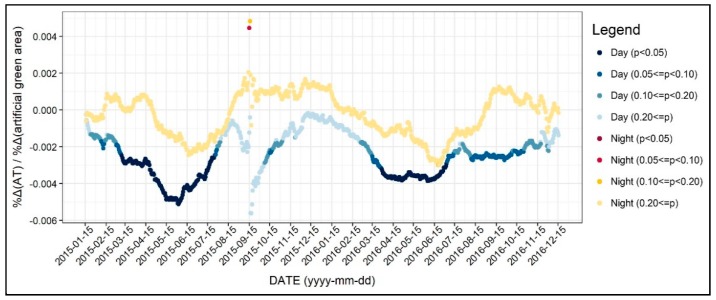
Changing patterns in the regression coefficients and their significances (artificial green area).

**Figure 16 ijerph-17-00421-f016:**
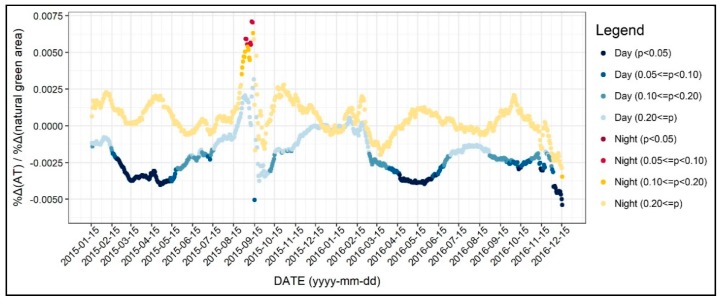
Changing patterns in the regression coefficients and their significances (natural green area).

**Figure 17 ijerph-17-00421-f017:**
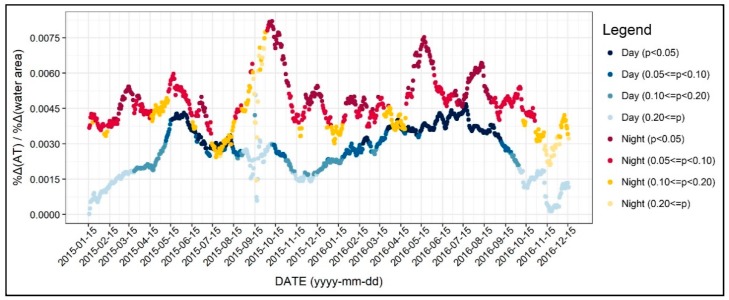
Changing patterns in the regression coefficients and their significances (water area).

**Table 1 ijerph-17-00421-t001:** Descriptive statistics.

Variable	Unit	Mean	St. Dev *	Min.	Max.	VIF **
AWS altitude [ALT]	m	63.136	33.001	9.727	332.600	1.56
Surface albedo [SA]	-	0.100	0.007	0.078	0.129	3.26
Residential use (floor area) [RA]	km^2^	0.585	0.347	0.000	1.489	3.81
Commercial use (floor area) [CA]	km^2^	0.107	0.106	0.000	0.637	2.36
Busi. & Industrial use (floor area) [BA]	km^2^	0.202	0.300	0.000	2.163	2.28
Traffic road area [TA]	km^2^	0.119	0.060	0.000	0.593	2.23
Building floor average [BF]	-	3.866	2.004	1.043	19.851	1.81
Area sky view factor [SVF]	-	0.630	0.124	0.368	0.999	4.47
Artificial green area [AG]	km^2^	0.049	0.051	0.000	0.350	1.74
Natural green area [NG]	km^2^	0.088	0.133	0.000	0.689	4.23
Water area [WA]	km^2^	0.013	0.042	0.000	0.406	3.02

Note: * Standard Deviation; ** VIF values were estimated for independent variables in the regression model.

**Table 2 ijerph-17-00421-t002:** Correlation matrix of independent variables.

Variable	ALT	SA	RA	CA	BA	TA	BF	SVF	AG	NG	WA
ALT	1.000										
SA	−0.010	1.000									
RA	−0.171	−0.475	1.000								
CA	−0.093	−0.288	0.090	1.000							
BA	−0.017	−0.254	−0.018	0.664	1.000						
TA	−0.291	−0.183	0.168	0.477	0.388	1.000					
BF	−0.121	−0.168	0.415	0.190	0.229	0.405	1.000				
SVF	0.159	0.479	−0.609	−0.354	−0.211	−0.070	0.120	1.000			
AG	−0.093	0.270	−0.230	−0.253	−0.182	0.114	0.209	0.550	1.000		
NG	0.526	0.246	−0.534	−0.410	−0.310	−0.570	−0.256	0.521	0.037	1.000	
WA	−0.216	−0.224	−0.176	−0.147	−0.122	0.100	0.158	0.396	0.325	−0.087	1.000

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
