# Peer review of "Diurnal and Seasonal Variations in the Effect of Urban Environmental Factors on Air Temperature: A Consecutive Regression Analysis Approach"

_ijerph, 2020, doi:10.3390/ijerph17020421_

Round 1

Reviewer 1 Report

The present article investigates the influence of different environmental factors on the temperature variations within the city of Seoul. A log-log regression between measured air temperature from a large network and other variables such as albedo, wind speed, land use etc. was conducted. The focus on the seasonality of the influences is an interesting aspect that is mostly disregarded because most studies focus on summertime. The article is well written and covers an important topic of the urban climate community. However, there are major and minor issues that need to be addressed before it can be published.

Major:

My main concern is the use of temperature observation 1,5 above rooftop level as a measure for urban temperatures. The authors claim that the air at rooftop level is mixed with the canyon temperatures and therefore representative for local-scale temperatures.  This however questionable because the mixing depends on meteorological condition, geometry of street canyons and buildings. Also, at 1.5 m the impact of the roof (e.g. albedo, heat release from the building) characteristics can be quite strong. One could argue that the time averaging might compensate for this spatial mismatch. But this would need to be proven with street canyon measurements and rooftop measurements.

Apart from the scale issue, the use of air temperature as a measure for heat stress in cities is not state of the art. The human comfort depends on more than just air temperature, e.g. radiation, moisture and wind speed. Hence, measures that reduce temperature e.g. increase of albedo might reduce the temperature but increase the radiation and therefore the mean radiant temperature (Schrijvers et al. 2016, Hoffmann et al. 2018). This has consequences for the conclusions and the recommendations for city planning.

Schrijvers, P. J. C., Jonker, H. J. J., de Roode, S. R., & Kenjereš, S. (2016). The effect of using a high-albedo material on the Universal Temperature Climate Index within a street canyon. Urban Climate, 17, 284-303.

Hoffmann, P., Fischereit, J., Heitmann, S., Schlünzen, K. H., & Gasser, I. (2018). Modeling exposure to heat stress with a simple urban model. Urban Science, 2(1), 9.

The review section is quite extensive and well structured. However, again there is mix of the definition of urban temperatures. Studies of urban air temperatures and urban land surface temperatures are mixed. The behaviour of both temperatures can be quite different (e.g. different diurnal cycle). Therefore, the impact of environmental factors can be different as well.

The scaling issue also relates to the independent variables such as the albedo, which is computed from LANDSAT 8 data. Due to the resolution of ~30 m this albedo is a mixture of albedos from roofs, streets, building walls, trees etc. It is not clear to me, how one can draw conclusions for city planners if it’s not clear which albedo needs to be changed.

With respect to statistical model it is not clear to me how the temperature – a variable that can be negative - normalized in order to compute the logarithm. This is also important for the interpretative of the results (beta = percentage of temperature change/percentage of independent variable change). I assume that the spatial temperature differences are used as dependent variable and not the absolute temperatures in °C. If so, the spatial variations in temperature can differ throughout the year. This would mean that the regression coefficients are not comparable between the different seasons (i.e. percentage changes do not correspond to the same absolute changes). This could explain the different magnitudes in summer and winter. Consequently, it would be good to compute and plot the spatial range or standard deviation in addition to mean temperature (Figure 2).

In addition, the intendent variables are not independent of each other e.g. building heights and commercial area cover. It would be good to compute the correlation between the independent variables and discuss of these could have an effect in the analysis of the regression coefficient.

The values in table 1 for the average land use types (residential area, natural green area etc.) add up to 1.163 km2 while the area of the 500 m buffer is 0.78 km2.  Hence, there is a mismatch, which could potentially lead to wrong results.

Minor:

Title of the Literature Review should be more specific. E.g. “literature review of observed environmental effects on urban temperatures”

How does this study relate to the study of Yi et al. (2018) who also constructed a statistical model for urban temperatures in Seoul based on surface characteristics like the ones used in the present study?

Yi, C.; Shin, Y.; Roh, J.-W. (2018) Development of an Urban High-Resolution Air Temperature Forecast System for Local Weather Information Services Based on Statistical Downscaling. Atmosphere 9, 164.

What is the difference between natural and artificial green? Would an urban Park with trees count as artificially, natural green or a mixture of both?

Throughout the text: Use another word for “minute” e.g. small

Lines 32-33: The urban heat island is usually not defined with a fixed threshold. I would define the UHI as positive temperature differences between urban and rural temperatures.

Line 78: What is meant by “building footprint”? Footprint with respect to what quantity?

Line 129: What is meant by “heat circulation”?

Line 144: I never heard of land use density. I think the obstacle density or building density is meant here.

Line 209: Please add a citation here.

Line 210: Could you give a range of values for the UHI in Seoul?

Line 213: Are these hourly, daily mean or daily max and min temperatures?

Line 239: Please state the time zone for the time values.

Line 243: What do you mean by “consistent” values? Constant values?

Lines 247-249: Is the wind speed average over the same period as the temperature?

Line 256: aerosols

Lines 294-299/Figure 2: Please state how the temperature in Figure 2 was computed. I assume that this is the spatially averaged temperature. In addition, the sentences Line 296-299 should go into the figure caption.

Lines 328-332/Figure 3: It is not clear what is shown here. Is this the Rsquared for the statistical model based on all independent variables.

Lines 349-350: The influence of the altitude on the temperature does not mean that the UHI increases with decreasing height. This is due to the temperature laps rate (in general temperature decreases with increasing height). The laps rate can be different in the different seasons. In winter the atmosphere is more stable and sometimes the laps rate is even reversed (temperature inversion). This could be the reason why the regression coefficient is low in wintertime!

Line 385: they are not identical by similar!

Lines 396-397: This is not consistent with the conclusions from the analysis of potential radiation and building height, which do not show a large impact on the temperature variations. Hence, shading cannot be used to explain the results.

Line 407-409: This is confusing. What is meant by absolute values of the coefficient? Why should a small coefficient still show that these land use categories are the main contributor of high air temperatures?

Lines 469-476/Figure 17: This result is counterintuitive. I would assume that water bodies warm the air during the night and cool during the day. Water bodies are usually cooler than the air temperature during the day and depending on water body type (lake, cannel, river, sea) warmer than the air temperature during the night.

Figures: Axis labeling should be bigger.

Figures 5-17: If possible, use the same scale for the y-axis. You mentioned that using the log-log regression the impacts of the different variables can be easily compared. Having different scales for every figure is not beneficial for the interpretation of the results!

Reviewer 2 Report

This article explored the diurnal and seasonal variations of the air temperature by conducting a series of log-log regression analyses with multiple variables, such as land-use, albedo, sky view factor, etc.

The research background, research questions, and methodology are clearly stated with appropriate visualizations. However, I am having a hard time understanding the output of the models and corresponding figures (Fig 5 - 17 ). Specifically, what are the values on the y-axis of all those figures? Is that the change of the air temperature divided by the change of the independent variables? Why choose such values? Why not use a standardized coefficient instead? For example, I am lost at line 342-344. Besides, how can you tell the change of the coefficients on figure 5?

Also, the authors stated that the VIF is not a problem. Is the spatial correlation a problem in this study? For example, are the air temperature or the errors of those regression models spatially autocorrelated? If yes, non-spatial regression may not be appropriate. 

Some other comments:

Line 56: What do you mean by evidence, could you be more specific? Although some limitations are discussed in the literature, the word 'evidence' is still too vague.  Line 60: what are the other sample numbers, could you list those studies?
In the literature review, one example is mentioned regarding biased samples, but not insufficient sample issues. Please provide full names for all abbreviations, such as SVF, AWS. Figure 1: The two green colors are too close to distinguish. Line 232: what is the reason for using the smoothing average?

Round 2

Reviewer 1 Report

While the reviewer addressed most of my comments to my satisfaction, there is one major issue and some minor issues that remain:

Major

I am still not satisfied with the log-log correlation. While I understand that this is used because it can be applied to variables with different range of values, it is still problematic to apply it to absolute temperature values that are also changing the standard deviation.

First, since the log-log regression gives a regression coefficient that only has the percentage changes as units the absolute value of the temperature is of importance for the method applied in this study. A 10% change at a level of about 20 °C (i.e. 2 °C) is much larger than at 5 °C (i.e. 0.5 °C). This can partially explain the high values for the regression coefficients in winter. Therefore, it doesn’t make sense to use the absolute temperature as a dependent variable in a log-log regression in a study that investigates the seasonal effects. In particular, the interpretation of the strength of the relationship between temperature and the other independent variables as a function of season is questionable using this method.

Second, also the standard deviation of the temperature variable changes over time. This has also an effect on the log-log regression coefficient. Again, this makes the comparison of the regression coefficient at different time of the year with different standard deviation problematic.

These issues might be solved by subtracting the absolute temperature values with the minimum temperature for each time 30-day period and afterwards normalize the temperature new time series with the standard deviation of the whole period. In doing so you have a non-zero time series with small values during the time of the year when the absolute temperature has small standard deviations and large values where the standard deviation is large.

Minor:

Please use the word “insignificant” only if the significance was tested.

“In detail, the average intensity of UHI in Seoul is approximately 1.8°C while it increases until 4.3°C” – Is 4.3 °C also an average value?

Instead of “laps rates” use “temperature laps rates”

I understand, why the author chose to have different ranges for the y-axes. While I am still favorable to use the same range in each figure, it should be at least mentioned in the text that the ranges are different in combination with an explanation on why the authors chose to do so.

Author Response

# Reviewer 1

Open Review

English language and style

( ) Extensive editing of English language and style required 
( ) Moderate English changes required 
(x) English language and style are fine/minor spell check required 
( ) I don't feel qualified to judge about the English language and style 

Yes

Can be improved

Must be improved

Not applicable

Does the introduction provide sufficient background and include all relevant references?

(x)

( )

( )

( )

Is the research design appropriate?

( )

( )

(x)

( )

Are the methods adequately described?

(x)

( )

( )

( )

Are the results clearly presented?

(x)

( )

( )

( )

Are the conclusions supported by the results?

( )

(x)

( )

( )

While the reviewer addressed most of my comments to my satisfaction, there is one major issue and some minor issues that remain:

Major

  1. I am still not satisfied with the log-log correlation. While I understand that this is used because it can be applied to variables with different range of values, it is still problematic to apply it to absolute temperature values that are also changing the standard deviation.

    First, since the log-log regression gives a regression coefficient that only has the percentage changes as units the absolute value of the temperature is of importance for the method applied in this study. A 10% change at a level of about 20 °C (i.e. 2 °C) is much larger than at 5 °C (i.e. 0.5 °C). This can partially explain the high values for the regression coefficients in winter. Therefore, it doesn’t make sense to use the absolute temperature as a dependent variable in a log-log regression in a study that investigates the seasonal effects. In particular, the interpretation of the strength of the relationship between temperature and the other independent variables as a function of season is questionable using this method.

    Second, also the standard deviation of the temperature variable changes over time. This has also an effect on the log-log regression coefficient. Again, this makes the comparison of the regression coefficient at different time of the year with different standard deviation problematic.

    These issues might be solved by subtracting the absolute temperature values with the minimum temperature for each time 30-day period and afterwards normalize the temperature new time series with the standard deviation of the whole period. In doing so you have a non-zero time series with small values during the time of the year when the absolute temperature has small standard deviations and large values where the standard deviation is large.

Response: Thank you for your comment. We agree with your concern that it may be problematic to apply the absolute temperature values to examine the determinants of air temperature in a seasonal perspective. Based on your suggestion, we tried to solve the issue by “subtracting the absolute temperature values with the minimum temperature for each time period and dividing it by the standard deviation value.” Although we changed our dependent variable, most of the analysis results were consistent compared to the previous manuscript. Additionally, we revised our manuscript when we had changes in our analysis results. Moreover, we also think that your suggestion on normalizing the dependent variable led us to attain reasonable results. After we changed the dependent variable, we were still able to observe obvious diurnal and seasonal patterns in the impact of environmental factors on air temperature. Based on the new analysis results, we revised our analysis results section (see pages 9 to 16).

See page 1 for the revised abstract regarding the changes in our dependent variable. “As our dependent variable, we employed the smoothed 31-day moving average of air temperatures, which we controlled the seasonal variation by normalizing the values observed from 247 automatic weather stations from 2015 to 2016.”

See page 6 for the explanation on our new dependent variable. “In particular, because the mean and standard deviation values of air temperature vary throughout the year, we normalized the dependent variable by subtracting each temperature values with the minimum temperature of each day. Also, we divided the values with the standard deviation value of each time period.”

Minor:

Please use the word “insignificant” only if the significance was tested.

Response: Thank you for your comment. We only used the word “insignificant” when we were explaining about the p-values of regression coefficient in our revised manuscript.

“In detail, the average intensity of UHI in Seoul is approximately 1.8°C while it increases until 4.3°C” – Is 4.3 °C also an average value?

Response: Thank you for your comment. We revised the sentence to avoid any possible misunderstanding.

See page 5. “The average intensity of UHI in Seoul is approximately 1.8°C while the maximum intensity is about 4.3°C [59].”

Instead of “laps rates” use “temperature laps rates”

Response: Thank you for your comment. We revised the word.

I understand, why the author chose to have different ranges for the y-axes. While I am still favorable to use the same range in each figure, it should be at least mentioned in the text that the ranges are different in combination with an explanation on why the authors chose to do so.

Response: Thank you for your comment. We addressed that the ranges of figures are different, and we explained the reason.

See page 9. “It should be noted that the range of the y-axis varies by each independent variable because the range of coefficients differ by variables.”

Reviewer 2 Report

Thanks for the revision.

Author Response

We have already addressed all comments. Thanks for your valuable comments.